# Implementation approaches for leprosy prevention with single-dose rifampicin: A support tool for decision making

Fleur ter Ellen[1], Kaat Tielens[1], Christine Fenenga[2], Liesbeth Mieras[3]*,
Anne Schoenmakers[3], Mohammad A. Arif[4], Nienke Veldhuijzen[5], Ruth Peters[1],
Eliane Ignotti[6], Christa Kasang[7], Benedict Quao[8], Peter Steinmann[9], Nand Lal Banstola[10],
Joshua Oraga[11], Teky Budiawan[12]

1 VU University, Amsterdam, the Netherlands, 2 Global Health Inclusive, Amsterdam, the Netherlands,
3 NLR, Amsterdam, the Netherlands, 4 Consultant, Delhi, India, 5 Leprosy Research Initiative, Amsterdam,
the Netherlands, 6 State University of Mato Grosso, Cuiabá, Brazil, 7 GLRA/DAHW, Wurzburg, Germany,
8 National Leprosy Control Programme, Ghana Health Service, Accra, Ghana, 9 Swiss Tropical and Public
Health Institute, Allschwil Switzerland, University of Basel, Basel, Switzerland, 10 NLR Nepal, Kathmandu,
Nepal, 11 IDEA, Nairobi, Kenya, 12 NLR Indonesia, Jakarta, Indonesia

* l.mieras@nlrinternational.org

pntd.0010792

University of Ceará, Fortaleza, Brazil, BRAZIL

**Data Availability Statement:** Data cannot be
shared publicly because the data are recordings of
interviews and publicly sharing would violate the

## Abstract

### Background

In the past 15 years, the decline in annually detected leprosy patients has stagnated. To
reduce the transmission of *Mycobacterium leprae*, the World Health Organization recom-
mends single-dose rifampicin (SDR) as post-exposure prophylaxis (PEP) for contacts of lep-
rosy patients. Various approaches to administer SDR-PEP have been piloted. However,
requirements and criteria to select the most suitable approach were missing. The aims of
this study were to develop an evidence-informed decision tool to support leprosy pro-
gramme managers in selecting an SDR-PEP implementation approach, and to assess its
user-friendliness among stakeholders without SDR-PEP experience.

### Methodology

The development process comprised two phases. First, a draft tool was developed based
on a literature review and semi-structured interviews with experts from various countries,
organisations and institutes. This led to: an overview of existing SDR-PEP approaches and
their characteristics; understanding the requirements and best circumstances for these
approaches; and, identification of relevant criteria to select an approach. In the second
phase the tool's usability and applicability was assessed, through interviews and a focus
group discussion with intended, inexperienced users; leprosy programme managers and
non-governmental organization (NGO) staff.

### Principal findings

Five SDR-PEP implementation approaches were identified. The levels of endemicity and
stigma, and the accessibility of an area were identified as most relevant criteria to select an

privacy of the interviewees. Researchers may request access to the data from Infolep at info@infolep.org.

**Funding:** The authors received no specific funding for this work.

**Competing interests:** The authors have declared that no competing interests exist.

approach. There was an information gap on cost-effectiveness, while successful implementation depends on availability of resources. Five basic requirements, irrespective of the approach, were identified: stakeholder support; availability of medication; compliant health system; trained health staff; and health education. Two added benefits of the tool were identified: its potential value for advocacy and for training.

## Conclusion

An evidence-informed SDR-PEP decision tool to support the selection of implementation approaches for leprosy prevention was developed. While the tool was evaluated by potential users, more research is needed to further improve the tool, especially health-economic studies, to ensure efficient and cost-effective implementation of SDR-PEP.

### Author summary

The chance of contacts of leprosy patients developing leprosy can be reduced by providing a single dose of rifampicin. The implementation of this type of post-exposure prophylaxis can be done in various ways. This study led to the development of the SDR-PEP decision support tool to select the most suitable approach. It was developed in two phases; first, a tool was drafted based on a literature review and expert interviews, this was followed by phase 2 in which interviews and a focus group discussion with intended users of the tool were held. Five SDR-PEP implementation approaches that have been developed so far were identified. Apart from the characteristics of these approaches, the tool lists five basic requirements for the successful implementation of any approach, and criteria that help to select the best approach in a given context. A flowchart supports the selection process. The study found that the tool can also be used for lobby and advocacy, to clarify SDR-PEP implementation and the choice for an approach, and in training on SDR-PEP implementation. Information about costs and cost-effectiveness of the approaches is limited. Further research will help to continue to improve the tool.

## Introduction

Leprosy is a widespread disease that remains endemic, mainly in low- and middle-income countries (LMICs). In 2019 a total of 202,185 new patients were diagnosed worldwide [1]. Leprosy is caused by an infection with *Mycobacterium leprae* (*M. leprae*). It is a disease with a long incubation period, with an average of two to five years elapsing between the moment of infection and the onset of clinical symptoms [2]. These include skin lesions with loss of sensation and nerve damage, which can lead to permanent sensory and motor function loss [2]. Furthermore, it can affect eyes, joints, lymph nodes and internal organs [3,4]. Eventually, leprosy can lead to permanent impairments, resulting in stigmatisation and social exclusion [5]. This underlines the importance of prevention, early diagnosis and treatment.

From the 1980s till the 2000s, a steady decline was observed in the prevalence of leprosy [6]. This was mainly due to the introduction of multidrug therapy (MDT) and shortening of the treatment period [2,6]. However, a very slowly decreasing trend is seen in the past 15 years [7]. This shows that the strategy aimed at early case detection and treatment was not sufficient to stop *M. leprae* transmission. It is assumed that transmission can already take place in the

subclinical period [8,9]. However, reliable tests to identify the presence of *M. leprae* in the subclinical period are missing. Close contacts of people with leprosy are at higher risk of infection and development of the disease than the general population [10–12]. Close contacts are usually defined as household members (sharing roof and kitchen), neighbours (i.e. living within around 100 metre radius) and other social contacts such as work colleagues or classmates, having been in contact with the patient for at least 20 hours per week for three months or more [13].

In 2008 the results of the epidemiological study on contact transmission and chemoprophylaxis in leprosy (COLEP) in Bangladesh were published. The trial established the effectiveness of post-exposure prophylaxis (PEP), in the form of a single-dose of rifampicin (SDR), for contacts of leprosy patients [14]. This study proved that SDR-PEP reduced the risk of leprosy among contacts by 57%. In a consequent multi-country feasibility study (the LPEP Program), the integration of SDR-PEP administration in routine leprosy control programmes was found to be feasible and acceptable by patients, contacts and healthcare workers [15–19]. Based on this, the World Health Organization (WHO) emphasizes in its "Guidelines for the Diagnosis, Treatment and Prevention of Leprosy", published in 2018, that transmission reduction can only be achieved through the combination of early case detection, timely treatment and chemoprophylaxis for screened and eligible contacts of leprosy patients [13,20]. In WHO's recently published Global Leprosy Strategy 2021–2030 "Towards Zero Leprosy" [21], preventive chemotherapy is recognized as showing promise for achieving the goal of interruption of leprosy transmission.

A feasible, acceptable and cost-effective implementation of contact screening and SDR-PEP administration within a leprosy control programme aligned with the local context is crucial for it to have a long-term impact. Various approaches to administer SDR-PEP have been piloted. Yet, to date no supporting tool to help decide which approach would be the best option for a given setting exists. This gap was identified during an international workshop on how to scale up SDR-PEP and advance research with representatives from Ministries of Health, non-governmental organizations (NGOs), universities and persons affected by leprosy [22]. To fill this gap, this study was set-up by several workshop participants, and involves some workshop participants as both experts and co-authors (n = 6).

The aim of this study was two-fold. Firstly, to develop a decision support tool for leprosy programme managers in endemic countries to select the most suitable SDR-PEP implementation approach for their setting. Secondly, this study aimed to get insights into the perceptions of stakeholders with limited or no SDR-PEP implementation experience on the usability and applicability of the SDR-PEP decision tool, to develop an improved tool that can contribute to the effective implementation of SDR-PEP.

## Methods

### Ethics statement

In line with the standards set by the Research Ethics Review Committee of the Faculty of Sciences, Vrije Universiteit Amsterdam, the Netherlands ethical approval was obtained for Phase 2 of the study (2021.016) [23]. The committee declared that the research proposal complied with the ethical guidelines of the faculty. Verbal consent was obtained and recorded from all interviewees during both study phases.

This study was conducted in two phases from April 2020 to June 2021: 1 –The development of the decision support tool; and 2 –the improvement of its usability and applicability (Fig 1). Phase 1 compromised a literature review combined with semi-structured expert interviews to build upon and supplement the findings from the literature review and design the decision

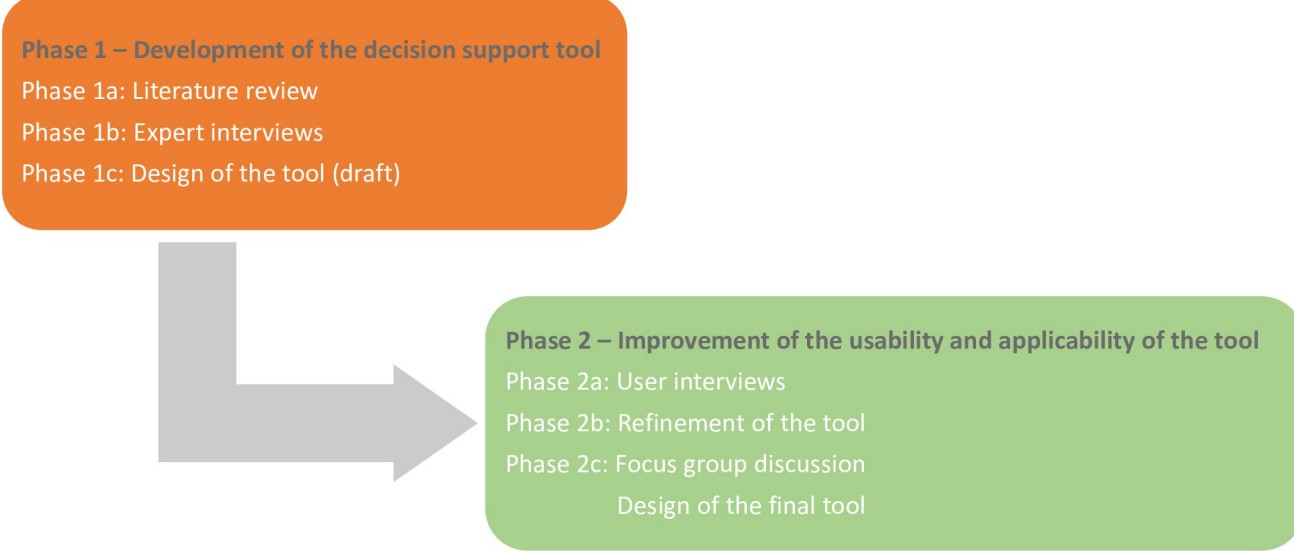

**Fig 1. Visual representation of methodology.**

support tool. Phase 2 was set up to evaluate the usability and applicability of the decision support tool with the intended users [24–27]. The second phase was an iterative process where the data of the first phase was used and the usability and applicability of the decision support tool was evaluated with the intended users. Two qualitative research methods (interviews and a focus group discussion [FGD]) with methodological triangulation as a validation strategy for the decision tool were used in Phase 2 (Fig 1).

## Phase 1: Development of the SDR-PEP decision support tool

**Literature review.** For the literature review, three databases were used: Web of Science, PubMed and Embase. The search string is presented in Box 1. The syntaxes used were made

---

Box 1. Search string

(leprosy OR leprae OR lepra OR Hansen OR hansens disease OR tuberculosis OR TB OR NTD OR neglected tropical disease) AND (preventive OR prophylaxis OR prophylactic OR PEP OR post-exposure prophylaxis OR SDR OR SDR-PEP PR single dose of rifampicin OR single dose rifampicin OR rifampicin) AND (blanket approach OR close contact approach OR screening OR hotspot blanket approach OR retrospective active case finding OR drives OR RACF OR urban approach OR skin camp approach OR skin camp OR stone-in-the-pond OR social network analysis) AND (contact tracing OR contact screening OR contact identification OR contact listing OR screening approach) AND (effects OR performance of tracing OR performance of screening OR staff OR accessibility OR cost-effectiveness OR efficacy OR efficiency OR stigma OR endemicity OR endemic OR type of contacts OR contacts OR barriers OR facilitators OR acceptability OR feasibility OR equity OR resources OR benefits OR harms OR challenges OR successes).

---

specific to fit the conventions of each database, but the content was kept the same for all. The search string was modified to search through titles and abstracts from articles published in the period 2005–2020. Articles were de-duplicated and a review was performed, first focusing on the abstracts and after provisional inclusion in the full texts. The aim of the literature review was to find SDR-PEP implementation approaches, their characteristics and requirements for implementation and to identify criteria to help select an approach. Exclusion criteria were: articles written in languages other than English; laboratory research; articles for which no full text could be obtained; articles that did not provide information on chemoprophylaxis methods, contact tracing and/or prevention of leprosy; rifampicin as part of other regimen; no information about approaches or no results; prediction models of future trends. This search took place in May and June 2020.

The review was performed by using a data extraction sheet, to summarise relevant aspects by one of the authors and reviewed by others [28]. Data on the following aspects were extracted: country; type of approach and characteristics; context; stated pros and cons; resources required; acceptability.

**Expert interviews.**   Semi-structured interviews were conducted with leprosy experts after consent was obtained. They were purposively selected based on experience with one or more of the SDR-PEP implementation approaches, either in research or in programme implementation. Selected experts were from multiple countries across different continents, diverse epidemiological settings, and working with various organisations (i.e. governments, WHO, NGOs, organisations for persons affected by leprosy, research institutes).

The aim of the interviews was to collect information on SDR-PEP implementation approaches, by reflecting on the approaches found in the literature and asking about experiences with those and other SDR-PEP implementation approaches. Experts were also asked about requirements for the implementation of SDR-PEP and criteria for the selection of the most suitable approach.

The interviews were recorded and transcribed by one of the authors. Thematic coding was done using Atlas.ti (Berlin: Atlas.ti Scientific Software Development GmbH). After coding the first two interviews an evaluation was done to determine whether the codes used were appropriate; a few additional codes were added. In addition, horizontal analysis was performed by comparing the interviews with each other. After analysis, findings were validated by sharing them with the interviewees, soliciting their approval on correctness and completeness of findings.

The data collected from the literature and the expert interviews fed the development of the tool, resulting in a first draft of the decision support tool.

## Phase 2: Improvement of the usability and applicability of the tool

In Phase 2, interviews and a FGD were conducted providing in-depth insights into the perceptions regarding the tool of stakeholders working on leprosy control and of persons affected by leprosy. Purposive sampling was used to select interviewees and participants, representing stakeholders working in leprosy control and persons affected by leprosy. They were different from the interviewees in Phase 1, because they were selected based on their lack of experience with SDR-PEP research or implementation and because they showed interest in starting SDR-PEP implementation.

After obtaining interviewee's verbal consent, interviews were recorded and transcribed verbatim by one of the authors. The interviewer used a semi-structured topic guide, that was refined through an iterative process [29]. This study captured both theoretical and inductive thematic analysis, as a codebook was developed before data analysis based on the initial

decision tool and theory, but themes also emerged from the data itself [30]. The focus group design was based on the first results of the data analysis of the interviews. The FGD was led by a moderator accompanied by an observer. The data from the FGD and user interviews resulted in a refined version of the tool.

## Results

### Phase 1 –Development of the SDR-PEP decision support tool

**Phase 1.a. Literature review.** A total of 467 publications were found; leaving 358 after deleting duplicates, of which 80 publications were considered relevant after scanning the titles and abstracts. The full texts of the 80 articles were carefully examined. Eight additional articles were added based on the suggestions of leprosy experts involved in the study and screening of the bibliographies of the relevant articles ('snowball method'). The full text reviews excluded 55 articles based on lack of relevant details or irrelevance for the subject of this study, resulting in 33 articles which were included in this review (Fig 2). The articles were published between 2005 and 2020 and included mainly operational studies related to active case finding and preventive treatment for leprosy and some other infectious diseases tuberculosis and/or leishmaniasis and/or other infectious diseases, n = 7).

**Phase 1.b. Expert interviews.** In Phase 1.b., 13 leprosy experts were interviewed, including experts working for an NGO, for the Ministry of Health (MoH), WHO, scientists and persons affected by leprosy. All participants of Phase 1.b. had been involved in SDR-PEP implementation in a wide variety of, mostly high endemic, countries. (Table 1). Six experts have co-authored this article.

The literature review and interviews with experts covered expertise with SDR-PEP implementation in 11 countries in Asia, Africa and South America. Based on the review and the interviews with experts in Phase 1 of the study, five types of SDR-PEP implementation approaches (A-E) were identified, plus two sub-types. Two approaches found in literature for

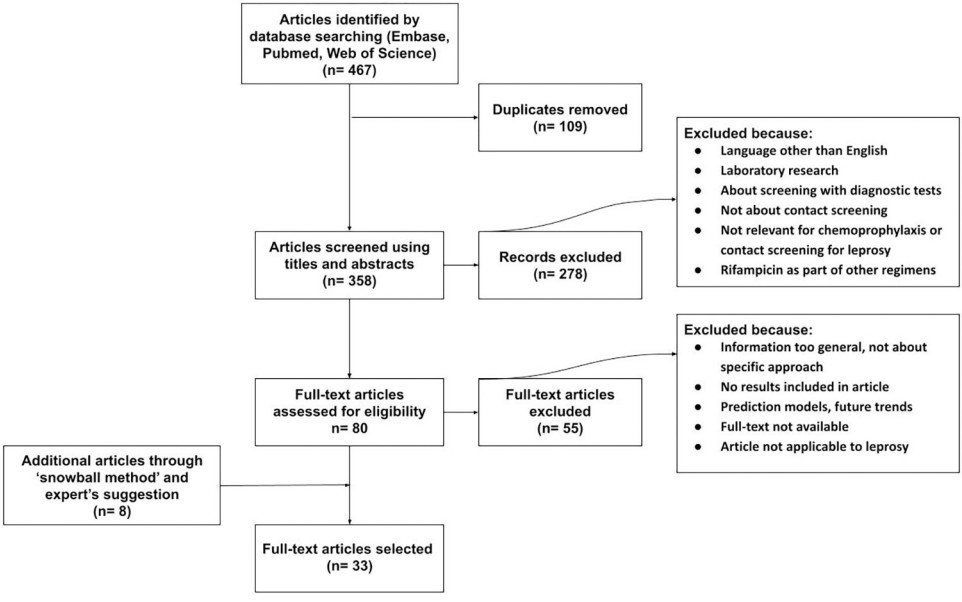

**Fig 2. Prisma flowchart of search results for literature review.**

**Table 1. Participants–Phase 1.**

| Participants Phase 1. b. (n = 13) | Affiliation | Countries |
|---|---|---|
| Semi-structured interviews (n = 13) | NGO staff (n = 6), MoH (n = 2), Persons affected by leprosy (n = 1), scientists (n = 3), WHO (n = 1) | Nepal (n = 3), Sri Lanka (n = 2), India (n = 2), Netherlands (n = 1), Switzerland (n = 1), Brazil (n = 1), Tanzania (n = 1), Indonesia (n = 1), Madagascar/Comoros (n = 1) |

contact screening and chemoprophylaxis provision for tuberculosis were excluded, because they depend on the outcome of lab screening tests for contacts, which are not currently available for leprosy. School screening also came up in the literature search; it is used for active screening to detect leprosy patients but has, according to the literature found and the experts, not yet been used for SDR-PEP implementation and was therefore excluded.

Table 2 provides an overview of the sources for the approaches identified and for the basic requirements for implementation. The characteristics of the approaches are described below and summarized in the SDR-PEP decision support tool page 3–5 (S1). Fig 3, shows a table with information found on the main characteristics of the different approaches (acceptance/ compliance, targeted contacts, cost-(effectiveness) and specific resources), and it indicates whether this information was found in literature, based on expert opinion or both (Fig 3 and S1).

### SDR-PEP implementation approaches.

A. *(standard) Close contact approach*

A.1. Close contact approach with disclosure of index patients' disease status:

In this approach, close contacts of newly identified leprosy patients are screened, after obtaining consent of the index patient. Contacts are given SDR-PEP when found eligible (in the absence of contraindications) [14,31]. This approach usually aims to reach 20 of the closest contacts, in line with the results of the COLEP trial. However, it is recommended, in literature as well as by the interviewees, not to adhere to the number of contacts rigidly, but to also take into account practical circumstances such as closeness of neighbouring houses, household size and health system characteristics to determine the number of contacts [32]. One expert said (I-R3, Male, NGO, interview): *"that magic number of 20 is not really based on evidence, (. . .) let the index case determine who those close contacts are (. . .)."*

Its use has mostly been described for high endemic settings, but Khoudri et al. described the close contact approach in Morocco, a low endemic country for leprosy [33]. The close contact approach can be implemented through house-to-house visits, or contacts of the leprosy patient can be requested to come to a public location or health facility. In several studies on screening of contacts of leprosy as well as tuberculosis patients, this has proven to be an acceptable approach for the main stakeholders involved (patients, their contacts and health workers) [16,32,34–38]. In the LPEP Program in which the close contact approach was used in five of the seven countries, around 1% of the leprosy patients refused to disclose their status and less than 1% of the contacts refused to receive SDR-PEP [16,19].

Idema et al. showed that even though implementation costs of the close contact approach differ per country and depend on endemicity, SDR-PEP is cost-effective for all contacts [39]. The standard approach applied in Dadra and Nagar Haveli, India, as part of the LPEP Program, was also found to be cost-effective in both the short and long term [40]. The experts did mention that door-to-door visits require a lot of time and resources, such as staff and transportation. However, during the LPEP Program in Sri Lanka only household contacts were included and they were invited to come to the health centre, which led to a low average

**Table 2. Sources used for the identification of approaches and basic requirements.**

| Approaches | Literature (n = 26) on leprosy, regarding (contact) screening and/ or chemoprophylaxis | Literature (n = 7) other infectious diseases, regarding (contact) screening and/or chemoprophylaxis | Expert opinion/ input (n = 13) +++ = most/all experts ++ = several experts + = one/few experts 0 = no experts | |
|---|---|---|---|---|
| Close contact approach | Moet et al.– 2008 (14) Apte et al.– 2019 (16) Richardus et al.– 2021 (19) Barth-Jaeggi et al.– 2016 (31) Steinmann et al. 2018 (32) Khoudri et al.– 2018 (33) Feenstra et al.– 2011 (35) Idema et al.– 2010 (39) Tiwari et al.– 2019 (40) | Volkmann et al.– 2016 (34) Martinez et al.– 2018 (36) Khatana et al.– 2019 (37) Morishita et al.– 2016 (38) Hanrahan et al.– 2019 (41) | +++ | A.1. |
| Non-disclosure approach | Richardus et al.– 2021 (19) Schoenmakers et al.– 2020 (42) | | ++ | A.2. |
| Self-screening approach | Steinmann et al.– 2018 (32) de Campos et al.– 2015 (43) Tiwari et al.– 2017 (BMC) (44) | | + | B. |
| Blanket approach | Tiwari et al.– 2017 (TMIH) (45) Tiwari et al.– 2018 (46) | | ++ | C.1. |
| fMDA approach | | | ++ | C.2. |
| Retro-active case finding approach | Cavaliero et al.– 2018 (47) Fürst et al.– 2018 (48) | | ++ | D |
| Skin camp / Community based approach | Schoenmakers et al.– 2021 (49) Msyamboza et al.– 2012 (50) | Banjara et al.– 2015 (51) Banjara et al.– 2019 (52) | + | E |
| **Basic requirements** | Smith et al.– 2014 (11) Bakker et al.– 2005 (12) Apte et al.– 2019 (16) Peters et al.– 2018 (17) Richardus et al.– 2021 (19) Steinmann et al.– 2018 (32) Feenstra et al.– 2011 (35) Tiwari et al.– 2017 (BMC) (44) Tiwari et al.– 2017 (TMIH) (45) Cavaliero et al.– 2018 (47) Ramasamy et al.– 2018 (53) Richardus et al.– 2018 (55) Da Cortela et al.– 2020 (57) Ortuno-Gutierrez et al.– 2019 (58) | | ++/+++ | |

number of contacts enrolled per index patient (n = 2), compared to the 'standard' of 20 [19]. A visit to the community for contact tracing and screening is evidently more time-consuming and more expensive than a facility-based approach but will lead to increased coverage [41].

A. 2. Non-disclosure close contact approach

A non-disclosure approach is mentioned by some experts as a novel approach. It is similar to the close contact approach but differs in that no disclosure of the disease status of the leprosy patient is required before contacts are traced and screened. Contacts are informed about leprosy having been identified in their geographical area, necessitating screening of community members and providing SDR-PEP to those who are found to be eligible [19,42]. The experts believed the non-disclosure approach to be very acceptable for leprosy patients, their contacts and health staff, and it would therefore be important to gain more experience with this approach.

## Characteristics of the approaches: obtained from literature/experts

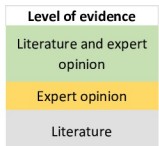

| Level of evidence |
|---|
| Literature and expert opinion |
| Expert opinion |
| Literature |

| Characteristics of the chosen SDR-PEP implementation approach(es): | | A. Close contact approach | | B. Self-screening approach | C. Blanket approach | | D. Retro active case finding (RACF) / Drives | E. Skin camp / Community based approach (pilots in initial stages) |
|---|---|---|---|---|---|---|---|---|
| | | A.1. (Standard) close contact approach | A.2. Non-disclosure approach | | C.1. Mass drug administration to entire population | C.2. Mass drug administration in clusters (not tested yet) | | |
| Acceptance of stakeholders | High / low | High | High | No information | High | Expected to be high | High | High |
| Targeted contacts | Household | Yes | Yes | Yes | Yes | Yes | Yes | Yes |
| | Neighbours | Yes | Yes | Yes | Yes | Yes | Yes | Yes |
| | Social contacts | Yes/No | Yes/No | Yes/No | Yes | Yes | Yes/No | Yes |
| | Community | No | Yes/No | Yes/No | Yes | Yes | No | Yes |
| Costs | High / low / cost-effective | Cost-effective for all contacts | | Low | High | No information | No information | High |
| Human resources | Medical staff / community health workers / volunteers | Medical staff required Community health workers and volunteers could help with screening | Medical staff required Community health workers and volunteers could help with screening | Medical staff required for confirmation or exclusion of leprosy | Medical staff required with support from community health workers | Medical staff required with support from community health workers | Medical staff required, leprologist or dermatologist if available | Medical staff required, preferably a dermatologist Community health workers and volunteers could help organising |
| Logistical preperations | Specifics for the approach | Door-to-door visits require more resources and preparations than a facility-based approach | | Self-screening forms containing pictures and text aids to self-check for signs of leprosy | A few resource intensive visits | Mapping, and geospatial analysis should be used to identify high endemic areas | Requires accurate register of leprosy patients | Requires good referral system |

**Fig 3. Characteristics of the approaches: obtained from literature/experts.**

B. *Extended contact-tracing with self-screening approach*

This approach is aimed at including a higher number of contacts by encouraging contacts to self-screen and screen their family members, using a form with instructions and illustrations to give guidance to the self-screening process [43]. People who self-screen as 'positive' are requested to consult a health worker to confirm or rule out leprosy [32,44]. All eligible contacts are provided with SDR-PEP. The aim of this approach is to make people more aware of signs and symptoms of leprosy and to include a higher number of contacts without increasing the workload of health workers.

The self-screening approach is designed to be less resource and labour intensive than the standard contact screening approach [32,44]. Although some experts indicated that in practice, experienced health staff were still requested to examine most contacts even after they self-screened, which is at odds with a reduced workload. Furthermore, limited involvement of health staff could also lead to leprosy patients being missed. It was suggested by the experts interviewed in this study that further research is needed to determine the specific acceptability, compliance, sensitivity and cost-effectiveness of this approach.

C.1. *Blanket approach / mass drug administration*

The blanket approach targets an entire community as in a mass drug administration campaign. All community members are screened and SDR-PEP is administered to all eligible persons in the community. Tiwari et al. advised a door-to-door strategy to minimise the number of people who may otherwise potentially be missed [45,46]. In their study, which was conducted in Lingat village on a remote island in Indonesia, no one refused consent, indicating

that the blanket approach was very well accepted in that setting. This was confirmed by the experts interviewed in this study mentioning high acceptance due to full community inclusion and involvement of opinion leaders in this approach, whereby the identity of the leprosy patients does not need to be disclosed.

The blanket approach performed in Indonesia as part of the LPEP Program demonstrated that a team of eight trained health staff could examine approximately 250 people per day [45,46]. This approach is rather resource intensive, requiring sufficient (community) health staff and a comprehensive logistic preparation, especially if it is implemented in hard-to-reach areas [46]. However, it requires only few but labour-intensive and longer visits to the community; once annually or bi-annually. This is a clear advantage of the blanket approach over a standard close contact approach, according to the experts, though others stressed that in case of large populations, the resources and time required would make the blanket approach less favourable.

### C.2. *Blanket approach / mass drug administration in clusters*

Mass drug administration in clusters ('hotspots' or 'pockets') or focal mass drug administration (fMDA) is a novel approach in leprosy control. It is similar to the blanket approach / mass drug administration (C1) in that it targets entire communities in which leprosy patients have been diagnosed. It is different in that it could be combined with self-screening and/or serology screening to avoid large numbers of community members having to be screened by health staff. The approach has not been piloted yet, but is seen as a promising, effective approach to reduce transmission at the community level in high endemic areas or so called 'hot-spots'. To prepare for fMDA, mapping and geospatial analysis should be used to identify clusters. Following this, the entire cluster is targeted.

As the fMDA approach has not been studied yet, little is known about its acceptability. However, the experts believe that since everyone in a particular area will be screened and disclosure of the patients' disease status is not needed, acceptability is likely to be high, similar to the blanket and non-disclosure approach, though the applicability and acceptability of serology screening in this context is not known. This approach will be resource intensive, comparable to C1.

### D. *Retro-Active Case Finding approach / Drive*

In Retro-Active Case Finding (RACF), also called 'Drives', contacts (household members and neighbours) of leprosy patients diagnosed in previous years are traced and screened [47,48]. SDR-PEP is administered to all contacts found to be eligible. According to the literature and the experts the retrospective period varies from 2 to 11 years. Fürst et al. described that the coverage of contacts of retrospective cases reduces the longer it has been since the index patient's diagnosis was made, because people may have moved or passed away. The RACF approach was reported to have a high level of acceptance among contacts, with only 2.7% refusing SDR-PEP [48]. Drives are performed by mobile teams with trained health staff and, where available, an experienced leprologist. This approach requires the availability of an accurate, historical database of leprosy patients and comprehensive logistic preparation. One interviewee mentioned that RACF cannot be considered a routine approach (I-R4, male, researcher, interview): "*It doesn't leave behind any system where people who see a patch or something can go and find an easy diagnosis, but it serves to diagnose patients actively.*"

### E. *Skin camp / Community based approach*

Skin camps are set up in a community, in which one or more leprosy patients have been identified and are executed by mobile teams with health staff, ideally including a

dermatologist, to screen people for skin diseases [49–51]. To increase coverage, pre-camp activities are to be organised to assemble as many people as possible. This integrated screening intervention can be combined with joint awareness-raising and community education to decrease stigma and is in line with the WHO road map for NTDs 2021–2030 [21]. Management of skin camps is logistically challenging and time-consuming, hence local health staff, volunteers and community leaders are often asked to help in organising the camps.

This approach does not require disclosure of the leprosy patient and enables covering a high number of community contacts per patient. An additional advantage is the diagnosis and treatment of other skin diseases. Skin camps seem to be well accepted by health staff and within communities [52]. However, there is no evidence yet on the acceptability of skin camps in which SDR-PEP is distributed. An acceptability study forms part of the PEP4LEP study which is currently being conducted in Ethiopia, Mozambique and Tanzania [49]. (I-R3, male, NGO, interview) *"If they come in large numbers and all they get is a note, which says that they have to buy the medicine themselves then the motivation [to attend the skin camp] probably will diminish."* However, free of charge SDR-PEP administration could possibly serve as an incentive, increasing community members' participation in the skin camps. Also, topical skin medication can be provided for other skin diseases.

Banjara et al. pointed out that costs of skin camps decrease when they become part of the routine operations of integrated programs [51]. Cost-effectiveness studies are needed to compare and understand the long-term effects of the skin camp approach compared to other approaches.

**Basic requirements for the successful implementation of any approach.** A set of basic requirements, considered important before any SDR-PEP approach can be implemented, was identified in the literature (Phase 1.a.) and discussed in more detail in the expert interviews (Phase 1.b.) [11,31,32,38,41,42,44,47,53]. The requirements are described below and summarized in the final decision tool (S1).

1. *Stakeholder support*

Governmental support, financial support, involvement of stakeholders and persons affected by leprosy, support from communities and technical support are essential for a successful and sustainable implementation of contact screening and SDR-PEP implementation. During the interviews in Phase 1.b., most experts indicated that additional scientific evidence of the effectiveness of the approaches will enhance governmental support (i.e. human resources, logistics, diagnostics, drugs) and this is regarded essential for successful implementation. WHO is considered to play a key role in convincing governments to embrace SDR-PEP and in providing the required technical guidance. NGOs in a country are perceived as catalysers, supporting governments in piloting and implementing SDR-PEP.

2. *Medication*

Resources should also include availability of sufficient MDT to treat leprosy patients as well as rifampicin for PEP. Some of the experts interviewed made clear that in some countries, procurement of loose rifampicin for prevention of leprosy was challenging. An arrangement of WHO integrating SDR-PEP supply in the already existing MDT donation and supply programme was mentioned as a possible, effective solution.

3. *Compliant health system*

There needs to be a thorough understanding of the structure and organisation of the health system, including the level of decentralisation and integration of programmes [54]. Detailed knowledge on the leprosy control programme is required, specifically on the way case finding

is done and whether contact screening is routinely performed. This information is important to determine how the SDR-PEP approach will be communicated, organized and implemented at the various administrative levels. According to Tiwari et al., SDR-PEP can be integrated into different systems [44]. Official registration of rifampicin as indication for leprosy PEP in national medical/pharmacotherapeutic guidelines is important for the process of purchasing rifampicin and embedding this approach in routine leprosy control activities and health worker trainings sessions. Health systems should have, or commit to setting up, surveillance systems that can (be adapted to) monitor, evaluate and share information about the progress of the contact screening and SDR-PEP administration [55]. The experts interviewed stressed the value of electronic databases and the importance of mapping of leprosy patients, which would enable targeted approaches.

### 4. *Trained staff*

Trained health staff and the involvement of persons affected by leprosy are seen as the backbone for successful SDR-PEP implementation and should therefore be included in every approach [19,56]. According to the experts, overall awareness and acceptance can be increased by involving persons affected especially in health education. (I-R8, male, NGO, interview) *"They encourage and support their peers practically and during psychological consequences of leprosy."* Insufficient training, and frequent transfers of previously trained government health staff were mentioned as negatively influencing the quality of implementation. There is a need for continuous development and maintenance of knowledge and skills of health staff to diagnose leprosy and to manage leprosy patients and their contacts [13].

Participation of volunteers and people affected by leprosy in the programme were mentioned not only because that has a positive effect on contact tracing but also because it will help manage the workload. Besides local health staff, volunteers are often involved in contact screening and SDR-PEP implementation. However, three experts stressed that volunteers can only be involved in screening, not in administration of SDR-PEP, because of the medical responsibility and confidentiality.

### 5. *Health education*

Health education needs to be part of the implementation of all of the approaches [17,18]. Close collaboration with organisations of persons affected by leprosy is also seen as essential for health education and awareness raising and to obtain a thorough understanding of the acceptability of the different approaches. Trained persons affected by leprosy and local staff can be involved in counselling to make people feel comfortable and encourage participation [11,13,47]. Also, local health staff and community volunteers are commonly trusted by the community and their involvement is therefore likely to improve compliance [16,35,57].

**Phase 1.c. Design of the decision support tool.** The first version of the decision support tool was developed on the basis of the literature review and expert interviews and had the form of a one-page matrix. The descriptions of the identified approaches were found to be aligned with the steps for SDR-PEP implementation as defined by WHO in the Technical guidance on contact tracing and post-exposure prophylaxis [13].

The most relevant criteria for the selection of the most suitable implementation approach were found to be: level of endemicity; level of stigma; and accessibility of an area. These are described in more detail in the Phase 2, in which these criteria were used as a basis for a flowchart to support the decision-making process.

**Table 3. Participants–Phase 2.**

| Participants Phase 2 (n = 15) | Affiliation | Countries |
|---|---|---|
| Semi-structured interviews (n = 14) | NGO staff (n = 5), MoH (n = 7), Persons affected by leprosy (n = 2) | Nigeria (n = 2), Congo (n = 4), Trinidad & Tobago (n = 2), Ghana (n = 2), St. Lucia (n = 1), Uganda (n = 1), Ivory Coast (n = 1), Kenya (n = 1) |
| Focus group discussion (n = 6) | NGO staff (n = 4), MoH (n = 2) | Nigeria (n = 2), Congo (n = 2), Ghana (n = 1), Vanuatu (n = 1) |

## Phase 2: Improvement of the usability and applicability of the tool

**Phase 2.a. User interviews.** To get insights into the perceptions of stakeholders with limited or no SDR-PEP implementation experience on the usability and applicability of the SDR-PEP decision tool, user interviews were conducted in Phase 2 of the study.

In total, 14 respondents were interviewed, 5 of which were also included in the FGD. One additional participant who has not been interviewed before also took part in the FGD. Participants of Phase 2 all worked in leprosy control in various countries either as NGO staff or for the MoH. Two members of an organisation representing persons affected by leprosy were included. All participants in Phase 2 had limited or no SDR-PEP implementation experience and were mainly from sub-Saharan African countries (Table 3). One interviewee co-authored this article. FtE conducted both the interviews and led the FGD.

A common view among interviewees was that the initial decision support tool was *"helpful"*, *"interesting"* and *"understandable"*. While observing and discussing the SDR-PEP decision support tool with the respondents in Phase 2, it became clear that the tool could be used for various purposes. While most respondents perceived the tool useful in selecting an appropriate SDR-PEP implementation approach, some felt that it could also help guiding the implementation process to train health workers or support decisions made regarding SDR-PEP implementation activities. A few respondents expressed that the tool helped them to understand the implementation approaches better and therefore tended to use it more as an informative document rather than a decision tool. For example, one respondent explained that the described characteristics of the different approaches were very helpful: *"Because even though I started with having a lot of information about SDR-PEP, sometimes I saw things that were new to me. So after I read this, I just understand"* (II-R2, male, NGO, interview). Also, one of the participants explained that the tool could be used to inform others on the decision that has been made: *"It will be very helpful in informing the person who is designing the project to show okay, this is the approach I will use"* (II-R1, male, NGO, interview). Five of the respondents reported that the tool could be used as a guideline to see what should be prepared in their country to start SDR-PEP implementation or to determine which steps ideally should be taken during the implementation process. The following comment illustrates this view: *"I first thought: 'Wow it makes things easy'. Because when you are thinking through the process of implementing, it is easy, you look at it and you know, this is what I am supposed to do. It makes planning easy"* (II-R4, female, MoH, interview).

**Phase 2.b. Refinement of the decision support tool.** Suggestions for improvements were also made, as a respondent mentioned: *"The tool as it is, is good for someone well-informed and very technical, so we need to make it more user-friendly, you know? Maybe step-wise"* (II-R8, male, MoH, interview). Also, some critical comments were made about the extent of information on one page. Respondents wanted the tool to provide a clear ranking for recommended

SDR-PEP implementation approaches according to the criteria. This encouraged us to revise the section on the recommended approaches and develop a flowchart to support the selection process. In agreement with respondents, the reorganization of the tool considered the level of endemicity; the level of stigma; the accessibility of an area; characteristics of the community (e.g. awareness of leprosy); the perceived sustainability of the intervention; and the available and required resources as the most important aspects to consider when selecting an appropriate SDR-PEP implementation approach. Based on these findings, the level of endemicity in an area was chosen as starting point of the flowchart, followed by the level of stigma and the accessibility of an area (S1). After observing the use of the tool in the FGD and discussing the initial decision support tool with the respondents, a differentiation was made between the logistical preparations and human resources required for the different SDR-PEP implementation approaches.

Findings related to the main selection criteria included in the final decision support tool are described below and were obtained from an iterative process in both study phases.

**Endemicity.**    Experts in Phase 1 discussed the importance of assessing the endemicity of leprosy in an area when selecting an SDR-PEP implementation approach. Also, over half of the respondents in Phase 2 prioritized the endemicity of leprosy as the most important aspect to consider when selecting a suitable approach for an area. The endemicity of an area in which SDR-PEP is implemented partly determines the intensity of the workload. A higher number of patients implies more contacts to be screened. In very high endemic areas (clusters or 'hot-spots' or 'pockets') or in areas where the entire community can be considered contacts because leprosy is so highly prevalent, it may be more practical to screen the whole population. A definition of different levels of endemicity is given in the tool.

**Stigma.**    Respondents in both phases of the study argued that leprosy-related stigma can be a complicating factor when tracing contacts and implementing SDR-PEP, if not taken into account when planning these activities. Refusal to take part in an SDR-PEP intervention can be triggered by stigma and misconceptions regarding leprosy and/or SDR-PEP [17]. Because of high levels of leprosy-related stigma, some programmes only included household contacts [19]. Studies showed that leprosy patients are usually more willing to disclose their disease status to their household contacts (to facilitate screening and preventive treatment) compared to neighbours or other more distant social contacts [19,35]. However, limiting the intervention to household contacts decreases the impact of SDR-PEP, especially in high endemic areas, because it reduces the coverage and household members benefit less from the effect of SDR-PEP compared to neighbours [12,58]. Several experts interviewed in Phase 1 mentioned that perceived and internalized stigma may cause patients to refuse to give permission to communicate with contacts. This can be reduced by proper counseling and involving family of persons affected by leprosy to encourage and support them. (I-R7, male, DPO, interview) *". . .and if [they] share [their] personal testimony (. . .) the people feel much more at ease, right, and they feel comfortable also."* Methods to assess the level of stigma are suggested in the tool.

**Accessibility of area and available resources.**    In case SDR-PEP is implemented in areas that are difficult to access, because of geographical features such as distance, difficult terrain, mountains, sea or lack of infrastructure, the intervention requires intensive logistics. Whether members of the community are requested to come to the health facility or whether a health providers' team makes house visits in the hard to reach area, careful preparation is needed including: informing the community, ensuring sufficient resources, including staff and transportation, and a planning based on realistic time allocation [32,45].

**Phase 2.c. FGD and design of the final decision support tool.**    During the FGD, two participants went through the flowchart with a geographical area in their country in mind where

SDR-PEP implementation was considered. Going through the flowchart enabled them to identify the most appropriate implementation approach for their area chosen, according to the tool. In the FGD, both the flowchart as well as the more in-depth matrix with characteristics of the approaches were shown. One participant indicated that many programme managers and staff working in leprosy control are not *"technically adept"* and therefore suggested to start with the flowchart for those who prefer not to delve further into the more complex matrix. This participant mentioned that, as part of capacity development, both the flowchart and the matrix should be part of the tool. Most participants agreed to this idea. Also, a suggestion was made to include supporting instructions on how to determine the level of endemicity and stigma in an area. This was addressed by adding two annexes to the decision support tool.

Early engagement of persons affected by leprosy and the community in the decision-making process to ensure sustainable implementation in a given area was already advised during user interviews. The FGD participants confirmed the importance of the early involvement of different stakeholders regarding the selection of a suitable SDR-PEP implementation approach; they also emphasized the importance of including (international) NGO's supporting leprosy work in the decision-making process. Besides the main purpose of the tool the FGD identified other reasons for its use: as an advocacy tool to target governments or external funders, as it explains why an implementation approach was chosen, it can support targeted resource mobilization requests, and it can be used in training on SDR-PEP implementation.

The final decision support tool consists of a description of the SDR-PEP implementation approaches, the basic requirements for the successful implementation of any approach, the selection criteria to determine the most suitable SDR-PEP implementation approach presented in a flowchart, and the characteristics of the SDR-PEP approaches obtained from literature/ experts, presented in a matrix (S1).

## Discussion

SDR-PEP, reducing the risk of developing leprosy in contacts of leprosy patients by 57%, is a promising innovative step towards stopping the transmission of *M. leprae*. There has been a growing interest from various countries to start with the implementation of SDR-PEP, a process for which the SDR-PEP decision support tool is designed. The flowchart clarifies which approaches are most suitable in different contexts. This is important because often, public health interventions that show to be effective in one setting may prove ineffective in another [59]. The tool also provides an overview of the evidence gaps that should be overcome by further research.

The output of the literature review was limited due to the formulation of the search string, because chemoprophylaxis for leprosy prevention is still relatively new and because the search was limited to articles published in English. This was addressed by thoroughly going through the bibliographies, searching in an additional database and experts' suggestions. Certain approaches have a richer (scientific) evidence base than others due to the fact they have been applied and studied more frequently, in different settings, under different circumstances and in multiple countries. Several still ongoing studies are expected to provide further insights into the pros and cons of the different approaches under varying circumstances [49,60]. Analysing the available evidence on the circumstances under which different SDR-PEP implementation approaches work best as well as capturing experts' views, provided a rich overview of characteristics of the different approaches and criteria to choose the most suitable approach. However, additional evidence is needed especially on the cost-effectiveness of SDR-PEP [40]. Preventive drugs for NTDs–including SDR-PEP for leprosy–are usually of relatively low cost [42,61]. Initial expenses for the purchase of medication and logistical costs (including the last

mile supply chain) have to be made to see return on investment (number of prevented patients and prevention of disabilities). In addition to the required finances there are various medication supply chain challenges which should be taken into account, such as: inaccurate record keeping and data analysis; insufficient stocks; logistic delays; inappropriate medication storage; damaged, expired or contaminated medication batches; and accountability issues [62].

Another identified area for further research is the acceptability and sensitivity of, as well as the compliance to, self-screening. The SDR-PEP decision support tool can be further improved by describing in more details which methods and tools can be used to assess the criteria in the tool (e.g. mapping, stigma assessment). To further validate and improve the decision support tool, qualitative studies, including observations of the tool usage and evaluations of the approach chosen after using the tool, could be done to analyse these decision-making processes in a real-life setting [63].

Targeted research can improve the knowledge base on the suitability of the different implementation approaches and further enrich the SDR-PEP decision tool. This kind of research is in line with the priorities that have been defined for leprosy research in recent years [64,65]. Studies looking at skin camps combined with SDR-PEP administration in for leprosy endemic communities have only recently started and studies on fDMA have yet to take place [49]. The relevance of moving towards adaptive decision tools (ADT) in NTD control was already mentioned by Booth and Clements, highlighting that such tools should take continuously changing circumstances and new insights into account [66]. Therefore, the SDR-PEP decision support tool will be a living document that will be adapted based on continuous learning and insights from new study results. It could also be adapted to implementation of immunoprophylaxis.

Besides the WHO 'Guidelines for the diagnosis, treatment and prevention of leprosy' from 2018 and the WHO technical guidance 'Leprosy/Hansen disease: Contact tracing and post-exposure prophylaxis' published in 2020, information on experiences with SDR-PEP implementation has been scattered and was not easily accessible for leprosy programme managers who want to embark on implementing SDR-PEP [5,13]. This study filled this gap by developing a decision support tool based on the currently available evidence and lessons learned. A next step is translating the decision tool into other languages. It is currently available in English via (S1).

## Conclusion

The SDR-PEP decision support tool provides a clear overview of the characteristics of five main SDR-PEP implementation approaches, five basic requirements for the successful implementation of any approach and several criteria that help to choose the most suitable approach in a given context (S1). A flowchart is inserted to support the selection process. The tool can also be used for lobby and advocacy, to clarify SDR-PEP implementation and the choice for an approach, and in training on SDR-PEP implementation.

More research is needed to further improve the tool, especially related to the cost-effectiveness of the various approaches and by defining the methods to assess the selection criteria mentioned in the tool.

## Supporting information

**S1 SDR-PEP decision support tool. SDR-PEP implementation approaches.**
(PDF)

## Acknowledgments

We are grateful to all participants that have contributed to this study.

## Author Contributions

**Conceptualization:** Fleur ter Ellen, Kaat Tielens, Christine Fenenga, Liesbeth Mieras, Ruth Peters, Christa Kasang, Peter Steinmann, Joshua Oraga.

**Data curation:** Fleur ter Ellen.

**Formal analysis:** Fleur ter Ellen, Kaat Tielens, Christine Fenenga, Liesbeth Mieras, Anne Schoenmakers, Nienke Veldhuijzen.

**Methodology:** Fleur ter Ellen, Kaat Tielens, Christine Fenenga, Liesbeth Mieras, Anne Schoenmakers, Ruth Peters, Christa Kasang.

**Supervision:** Christine Fenenga, Liesbeth Mieras, Anne Schoenmakers, Ruth Peters.

**Validation:** Mohammad A. Arif, Eliane Ignotti, Christa Kasang, Benedict Quao, Peter Steinmann, Nand Lal Banstola, Joshua Oraga, Teky Budiawan.

**Writing – original draft:** Fleur ter Ellen, Kaat Tielens, Christine Fenenga, Liesbeth Mieras.

**Writing – review & editing:** Fleur ter Ellen, Kaat Tielens, Christine Fenenga, Liesbeth Mieras, Anne Schoenmakers, Mohammad A. Arif, Nienke Veldhuijzen, Ruth Peters, Eliane Ignotti, Christa Kasang, Benedict Quao, Peter Steinmann, Nand Lal Banstola, Joshua Oraga, Teky Budiawan.

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
