## [Decision Letter · Decision Letter 0]

6 Apr 2022

Dear Dr Mieras,

Thank you very much for submitting your manuscript "Implementation approaches for leprosy prevention with single-dose rifampicin: a support tool for decision making" for consideration at PLOS Neglected Tropical Diseases. As with all papers reviewed by the journal, your manuscript was reviewed by members of the editorial board and by several independent reviewers. In light of the reviews (below this email), we would like to invite the resubmission of a significantly-revised version that takes into account the reviewers' comments. 

In addition to the recommendations and suggestions indicated by the reviewers, concern arose regarding the “Decision tool SDR-PEP implementation” tool, since it is available on the website of the organization Infolep – Leprosy information services, as per the link below: https://www.leprosy-information.org/resource/decision-tool-sdr-pep-implementation. Please clarify this situation.

We cannot make any decision about publication until we have seen the revised manuscript and your response to the reviewers' comments. Your revised manuscript is also likely to be sent to reviewers for further evaluation.

Sincerely,

Alberto Novaes Ramos Jr

Associate Editor

Epco Hasker

Deputy Editor

In addition to the recommendations and suggestions indicated by the reviewers, concern arose regarding the “Decision tool SDR-PEP implementation” tool, since it is available on the website of the organization Infolep – Leprosy information services, as per the link below: https://www.leprosy-information.org/resource/decision-tool-sdr-pep-implementation. Please clarify this situation.

Reviewer's Responses to Questions

**Key Review Criteria Required for Acceptance?**

**Methods**

-Are the objectives of the study clearly articulated with a clear testable hypothesis stated?

-Is the study design appropriate to address the stated objectives?

-Is the population clearly described and appropriate for the hypothesis being tested?

-Is the sample size sufficient to ensure adequate power to address the hypothesis being tested?

-Were correct statistical analysis used to support conclusions?

-Are there concerns about ethical or regulatory requirements being met?

Reviewer #1: This study is based on eliciting opinions about how best to implement post-exposure prophylaxis (PEP) as a preventive measure against leprosy.

Two groups of interviewees are described, one consisting of 13 individuals with prior knowledge of PEP implementation and the other of 15 individuals with little or no previous experience of PEP.

There are 15 authors of the paper, but their roles are not clearly described, although in the text 6 of the authors are said to have undertaken certain tasks.

In my opinion, there is a need for greater clarity about who did what in this study. What were the roles of each of the co-authors? How were the two groups of participants selected? Who carried out the interviews and FGDs? Were any of the authrs included in the list of experts?

There is very little information about exactly how the 5 types of PEP implementation were identified - it was simply "based on the review and expert interviews". One might imagine that certain tentative conclusions were drawn after the literature review, and the list was then refined through the interviews; more transparency about this process would be welcome.

Reviewer #2: The limitation of articles consulted is quite understandable, since this WHO recommendation is still not very widespread. 

This restriction may also have influenced the health professionals selection to be interviewed

However, the tool created should be useful to add-on WHO guideline for leprosy contacts tracing in SDR-PEP.

Reviewer #3: I consider this an important and timely publication in support of the implementation of novel preventive interventions in the field of leprosy. It is very well written and provides a wealth of information about the state of the art of SDR knowledge and experience, as well as very detailed and practical guidelines (the support tool) for deciding on appropriate strategies in given settings.

The publication is based on a mixed methods approach, including literature review, expert interviews and focus group discussions. I particular like the classification of close contact survey in various alternative approaches, which on one hand clarifies the different possibilities and certainly will help field programs to choose the most appropriate and feasible approach in their own setting. The description of basic requirements for implementation is also very informative and useful.

The focus of this publication is on the decision support tool that is based on the studies. A strong feature of this paper is that it includes the development and evaluation of the tool with the intended end users. In that sense it represents a full cycle of research, development, evaluation and presentation of the final product; the tool.

With regard to methodology, the only comment I have is that the literature review includes English language publications only. It may have been useful to include Portuguese, Spanish and French language publications as well? On the other hand, the interviews and focus groups included many people from endemic countries.

Reviewer #4: The study objectives are clear but no hypothesis was defined. It is important to define the hypothesis or the group assumption when developing this decision support tool about the SDR-PEP as well as evaluate its applicability.

Regarding the study design, the methodology is considered appropriate to support the defined goals.

The study population is clearly described, but I suggest moving part of the text that is placed in the results section as “Participants interviews and focus group” where this population had been better characterized, to the methods section. Also, in the methods section, in the paragraph that the participants of the second stage of the study are described, I suggest explaining more carefully that one of the participants was included only from the focused group.

About the compliance with ethical requirements, the number of the research ethics committee approval document may be inserted.

Reviewer #5: - The author clearly define the situation related the topic in the introduction but there is statement that confusing in Alinea 3..” this study proved… 57%’.. this statements should not appear in the introduction.

- The number 57% also did not support data in the result of the manuscript.

- Please arrange the introduction section so that easy to understand, the author, one paragraph one idea or topic

- The study consisted two phases, the first the development process with the literature review, the second was evaluated the usability and applicability with FGD and interview. 

- The first phase was described the query method and expert interviews. In the expert interviews authors were not clearly define how many experts, who etc. but it put in the result, while in the results are not given anything as the result of the process, same with the literature review in the result.

- If the process was complex, please put a diagram to describe it.

**Results**

-Does the analysis presented match the analysis plan?

-Are the results clearly and completely presented?

-Are the figures (Tables, Images) of sufficient quality for clarity?

Reviewer #1: Greater transparency in how the list of five types was generated is needed.

Reviewer #2: The authors statement related to the main limitations of the study are satisfactory and the discussion adequately connected the research objectives.

Reviewer #3: The results are clearly presented. In particular the tool itself, the main result, is very clear and accessible, with much detail.

Reviewer #4: In the results section – literature review, check the information related to the correct number of articles excluded after reading the full text. The flowchart states that 55 articles were excluded, which is consistent with the final number included in the review (33). However in the text body 52 articles are shown.

In the literature review, the main information extracted from the selected articles as well as the summarized data of these studies were not presented. I suggest the inclusion of a summary table as supplementary material. Throughout the text it is described that the data were extracted from the articles using a data extraction form. It defines what kind of information was cited such as country, type of approach and characteristics, context, stated pros and cons, resources required, and acceptability. However it does not show the extracted information, nor to which articles each information corresponds.

In addition, in the part that reports the interviews with experts, it was described that the collected information from the interviews were also used to develop the tool, but it does not make it clear which data were collected during the interview and which ones were important in its development. Besides, what are the guiding questions of these interviews? It is also necessary to explain the technique used to construct the five categories.

Some results, such as the one presented in the item “C.2. Blanket approach / mass drug administration in clusters” do not refer to any study in the literature review nor to the speech of any specialist who supports the definition of this category as presented for the other categories. Thus, it difficult to understand the emergence of this category among the five approaches considered to be the main ones in the formulation of the tool.

In general, the results have already been presented grouped in the approaches defined in the tool, mixing what was found in the literature with the speech of the specialists. Greater clarity of these data is important, as they support the conclusion of the study.

Reviewer #5: - Please arrange the results in way the author described in the methods consist of phase 1 an phase 2.

- The reader can not find the results of the study easily such as what is the result from literature review? Expert interview? And so on.. its confusing

- Add some tables as conclusion from all of the result in each phase don’t mix the result of each phase

- Add table result from: literature review, FGD and interview with thematic analysis 

- The reader difficult to find the meaningful result. The author should put the result in the manuscript not only link. Please choose the most meaningful result.

- Do not put any references in the results. just describe the result in each phase clear and easy to understand even though the study was conducted qualitative methods in collecting data.

**Conclusions**

-Are the conclusions supported by the data presented?

-Are the limitations of analysis clearly described?

-Do the authors discuss how these data can be helpful to advance our understanding of the topic under study?

-Is public health relevance addressed?

Reviewer #1: The final list is helpful.

Reviewer #2: Yes, it is relevant in addressing a current WHO recommendation and fills in the lack of studies on this subject

Reviewer #3: The conclusions are to the point. It also points out areas for further development of the tool. Costing aspects and cost-effectiveness analysis of SDR-implementation approaches is certainly a next step.

Reviewer #4: The study conclusion is clear and supported by the data presented throughout the article and in the document “Decision tool SDR-PEP implementation”, highlighting the importance of the developed tool. In addition, it responds to the proposed objectives. The limitation of the developed tool was described, as well as the need for new studies that could overcome its limitations as new analyzes that would result in the improvement of the appliance.

Reviewer #5: - Please do not discuss anything that not in the result. Author mention percentage numbers that not related in the manuscript.

- In the manuscript should give the details in SDR-PEP decision support tool so easy to discuss each component in it.

- Please arrange the discussion related the important result and component in SDR-PEP decision support tool

**Editorial and Data Presentation Modifications?**

Reviewer #1: (No Response)

Reviewer #2: There is still some resistance to the implementation of Chemoprophylaxis in the leprosy control programs. Therefore, perhaps it could be minimized. One suggestion is to highlight that the tool is useful not only for the SDR-PEP purpose, but also for immunoprophylaxis and early diagnosis (in the conclusion and in the abstract).

Reviewer #3: Well written report and very relevant for the field of leprosy. My advice is to accept.

Reviewer #4: (No Response)

Reviewer #5: The topic is interesting and important even though the manuscript should arrange to become readable

**Summary and General Comments**

Reviewer #1: This is a practical paper, describing five ways in which PEP could be implemented in a leprosy control program.

The list is very sensible, but many of the authors could probably have come up with the same list without having to do the study. So the authors need to show that their work had an element of rigour - that the reults were not actually decided beforehand; this woud be mainly demonstrated by a more detailed presentation of the results of the literature review and how those initial conclusions were refined through expert interviews.

The words "unexperienced" and "capacitated" may or may not be real words in English but they are certainly inelegant and need to be changed.

Reviewer #2: Although not really an innovation, the tool resulting from the study offers a hands-on systematization to be applied on contacts tracing and so that favoring an early diagnosis and leprosy prevention.

Reviewer #3: I think this is very well written and very useful for the field of leprosy. It not just synthesizes much knowledge, but also continues to provide a valuable tool to support implementation. A great example of well-conducted applied research!

Reviewer #4: A strong point of the study is that the developed tool will have wide applicability in clinical practice, making it possible to generate a positive impact on the formulation of leprosy prevention policies.

Reviewer #5: The manuscript is to abstract and missing the important things that should describe, the author should include the important result in the manuscript not to ask the reader to open many links

PLOS authors have the option to publish the peer review history of their article (what does this mean?). If published, this will include your full peer review and any attached files.

Reviewer #1: Yes: Paul Saunderson

Reviewer #2: Yes: Maria Leide Wand Del Rey de Oliveira

Reviewer #3: No

Reviewer #4: No

Reviewer #5: No
---

## [Decision Letter · Decision Letter 1]

29 Jun 2022

Dear Dr Mieras,

Thank you very much for submitting your manuscript "Implementation approaches for leprosy prevention with single-dose rifampicin: a support tool for decision making" for consideration at PLOS Neglected Tropical Diseases. As with all papers reviewed by the journal, your manuscript was reviewed by members of the editorial board and by several independent reviewers. The reviewers appreciated the attention to an important topic. Based on the reviews, we are likely to accept this manuscript for publication, providing that you modify the manuscript according to the review recommendations. 

Sincerely,

Alberto Novaes Ramos Jr

Associate Editor

Epco Hasker

Deputy Editor

Reviewer's Responses to Questions

**Key Review Criteria Required for Acceptance?**

**Methods**

-Are the objectives of the study clearly articulated with a clear testable hypothesis stated?

-Is the study design appropriate to address the stated objectives?

-Is the population clearly described and appropriate for the hypothesis being tested?

-Is the sample size sufficient to ensure adequate power to address the hypothesis being tested?

-Were correct statistical analysis used to support conclusions?

-Are there concerns about ethical or regulatory requirements being met?

Reviewer #1: The Methodology has now been explained clearly.

Reviewer #3: The authors have answered all reviewer questions meticulously and clarified the methods section considerably.

Reviewer #4: Thank you for considering the suggested changes and justifying those that were not met. I have no further considerations on this topic.

Reviewer #5: - In Phase 1. Literature review, please describe the search timeline of articles published (is it May to June 2020?) (page 7) because in the result the author mentioned that articles were published between 2005-2020 (page 10)

**Results**

-Does the analysis presented match the analysis plan?

-Are the results clearly and completely presented?

-Are the figures (Tables, Images) of sufficient quality for clarity?

Reviewer #1: The Results are comprehensive and clearly presented.

Reviewer #3: Good improvements after reviewer comments.

Reviewer #4: Regarding the comment "In the results section – literature review, check the information regarding the correct number of articles excluded after reading the full text. The flowchart states that 55 articles were excluded, which is consistent with the final number included in the review (33), and the body of the text contains 52." performed in the review of the original submission, I believe that I may not have been clear about the divergence found. In this way, I reiterate the comment since I consider it important to review the data to avoid discrepancies in your text.

I did not refer to the greater number of articles used in the article's reference list in relation to those used in the literature review but to the divergence between the information on the number of articles excluded from the literature review in the full-text reading phase, which n = 55 in figure 2 and n = 52 in the text description in the 5th line of the topic Phase 1 a Literature review. Please check this data and make the necessary corrections.

Reviewer #5: - The result is more readable than the previous version.

- Please do not put any comments, interpretations or references in the result, put them into the discussion (check page 14-27)

- Where I and the reader can find the SDR-PEP decision support tool?

**Conclusions**

-Are the conclusions supported by the data presented?

-Are the limitations of analysis clearly described?

-Do the authors discuss how these data can be helpful to advance our understanding of the topic under study?

-Is public health relevance addressed?

Reviewer #1: Fully based on the Results presented.

Reviewer #3: All well-described and improved based on reviewer comments.

Reviewer #4: I consider that the new topic structure meets the proposed changes. I have no additional considerations on this topic.

Reviewer #5: Conclusion clear

**Editorial and Data Presentation Modifications?**

Reviewer #1: None.

Reviewer #3: None suggested.

Reviewer #4: I consider that the new text structure meets the proposed changes. I have no additional considerations on this topic.

Reviewer #5: readable

**Summary and General Comments**

Reviewer #1: The revised version is greatly improved.

Reviewer #3: I am happy with the improvements made based on all reviewer comments.

Reviewer #4: The main recommendations made in the original submission were met and/or justified by the authors. I reiterate just one comment referring to the results section. It does not detract from the merit of the manuscript, but I consider it important to be revised, thus avoiding divergence between figure 2 and the writing of the topic "Phase 1.a. Literature review."

I would like to point out the need to remove the hyperlink setting throughout the text, in places where it previously contained a link and was replaced by the reference to the tool (see SDR-PEP decision support tool). This change is necessary since even though the page has been removed, the text remains with the configuration to redirect to the page in question (https://www.leprosy-information.org/resource/decision-tool-sdr-pep -implementation).

The changes made in this revision version made the text much clearer, which I believe will be useful in the adherence of professionals to using the proposed tool "Decision tool SDR-PEP implementation" since it has become easier to understand the process by which it was created, offering greater credibility to the tool.

Reviewer #5: - The paper is more readable than the previous version, the author already put the additional table, and figures to make the article more readable.

- It is still confusing between the result and discussion, the author mixed it

- Please provide the tool in the manuscript or link because it can not find in the link or in pdf

PLOS authors have the option to publish the peer review history of their article (what does this mean?). If published, this will include your full peer review and any attached files.

Reviewer #1: Yes: Paul Saunderson

Reviewer #3: No

Reviewer #4: No

Reviewer #5: No

Figure Files:

Data Requirements:

Reproducibility:

References

---

## [Decision Letter · Decision Letter 2]

2 Aug 2022

Dear Dr Mieras,

Thank you very much for submitting your manuscript "Implementation approaches for leprosy prevention with single-dose rifampicin: a support tool for decision making" for consideration at PLOS Neglected Tropical Diseases. As with all papers reviewed by the journal, your manuscript was reviewed by members of the editorial board and by several independent reviewers. The reviewers appreciated the attention to an important topic. Based on the reviews, we are likely to accept this manuscript for publication, providing that you modify the manuscript according to the review recommendations. 

Sincerely,

Alberto Novaes Ramos Jr

Academic Editor

Epco Hasker

Section Editor

Reviewer's Responses to Questions

**Key Review Criteria Required for Acceptance?**

**Methods**

-Are the objectives of the study clearly articulated with a clear testable hypothesis stated?

-Is the study design appropriate to address the stated objectives?

-Is the population clearly described and appropriate for the hypothesis being tested?

-Is the sample size sufficient to ensure adequate power to address the hypothesis being tested?

-Were correct statistical analysis used to support conclusions?

-Are there concerns about ethical or regulatory requirements being met?

Reviewer #1: The Methods are clearly explained.

Reviewer #4: The methodology is clearly described.

Reviewer #5: - In Phase 1. Literature review, please describe the search timeline of articles published (is it May to June 2020?) (page 7) because in the result the author mentioned that articles were published between 2005-2020 (page 10)  please define in Phase 1: the time period the search was performed and the articles published

**Results**

-Does the analysis presented match the analysis plan?

-Are the results clearly and completely presented?

-Are the figures (Tables, Images) of sufficient quality for clarity?

Reviewer #1: The Results are clearly explained.

Reviewer #4: Thank you for considering the suggested changes. The results are now clearly described.

Reviewer #5: - In Phase 2, the results are not from literature review but from user interview but author still put references in the results (page 26) so it’s mix between results from respondents and references.

**Conclusions**

-Are the conclusions supported by the data presented?

-Are the limitations of analysis clearly described?

-Do the authors discuss how these data can be helpful to advance our understanding of the topic under study?

-Is public health relevance addressed?

Reviewer #1: The Conclusions are fully justified.

Reviewer #4: The conclusion is clear and described based on the results.

Reviewer #5: - Do not mix between result and discussion

**Editorial and Data Presentation Modifications?**

Reviewer #1: None needed.

Reviewer #4: None suggested.

Reviewer #5: none

**Summary and General Comments**

Reviewer #1: A comprehensive and much apprecited paper.

Reviewer #4: The authors made the suggested changes, which made this version much clearer and improved. I have no additional suggestions for changes to the article.

Reviewer #5: - I hope the reader could easily access the tool to make comprehensive view. In the future the tool should be access widely to make useful.

PLOS authors have the option to publish the peer review history of their article (what does this mean?). If published, this will include your full peer review and any attached files.

Reviewer #1: Yes: Paul Saunderson

Reviewer #4: No

Reviewer #5: No

Figure Files:

Data Requirements:

Reproducibility:

References

---

## [Decision Letter · Decision Letter 3]

6 Sep 2022

Dear Dr Mieras,

We are pleased to inform you that your manuscript 'Implementation approaches for leprosy prevention with single-dose rifampicin: a support tool for decision making' has been provisionally accepted for publication in PLOS Neglected Tropical Diseases.

Best regards,

Alberto Novaes Ramos Jr

Academic Editor

Epco Hasker

Section Editor

Reviewer's Responses to Questions

**Key Review Criteria Required for Acceptance?**

**Methods**

-Are the objectives of the study clearly articulated with a clear testable hypothesis stated?

-Is the study design appropriate to address the stated objectives?

-Is the population clearly described and appropriate for the hypothesis being tested?

-Is the sample size sufficient to ensure adequate power to address the hypothesis being tested?

-Were correct statistical analysis used to support conclusions?

-Are there concerns about ethical or regulatory requirements being met?

Reviewer #1: Clear

Reviewer #4: The methodology is clearly described. I have no further considerations on this topic.

Reviewer #5: None suggested.

**Results**

-Does the analysis presented match the analysis plan?

-Are the results clearly and completely presented?

-Are the figures (Tables, Images) of sufficient quality for clarity?

Reviewer #1: Clear

Reviewer #4: The results are clearly described. I have no further considerations on this topic.

Reviewer #5: None suggested.

**Conclusions**

-Are the conclusions supported by the data presented?

-Are the limitations of analysis clearly described?

-Do the authors discuss how these data can be helpful to advance our understanding of the topic under study?

-Is public health relevance addressed?

Reviewer #1: Clear

Reviewer #4: The conclusion is clearly described. I have no further considerations on this topic.

Reviewer #5: None suggested.

**Editorial and Data Presentation Modifications?**

Reviewer #1: None

Reviewer #4: None suggested.

Reviewer #5: None suggested.

**Summary and General Comments**

Reviewer #1: Publication should go ahead without further delay.

Reviewer #4: The authors made all the changes suggested. I have no additional suggestions for the article.

Reviewer #5: None suggested.

PLOS authors have the option to publish the peer review history of their article (what does this mean?). If published, this will include your full peer review and any attached files.

Reviewer #1: **Yes: **Paul Saunderson

Reviewer #4: No

Reviewer #5: No

---

## [Editor Report · Acceptance letter]

7 Oct 2022

Dear Dr Mieras,

We are delighted to inform you that your manuscript, "Implementation approaches for leprosy prevention with single-dose rifampicin: a support tool for decision making," has been formally accepted for publication in PLOS Neglected Tropical Diseases.

Best regards,

Shaden Kamhawi

co-Editor-in-Chief

Paul Brindley

co-Editor-in-Chief
